# Recent Studies on the Development of Nicotine Abuse and Behavioral Changes Induced by Chronic Stress Depending on Gender

**DOI:** 10.3390/brainsci13010121

**Published:** 2023-01-10

**Authors:** Karolina Grabowska, Wojciech Ziemichód, Grażyna Biała

**Affiliations:** Chair and Department of Pharmacology with Pharmacodynamics, Medical University of Lublin, Chodźki 4A Street, 20-400 Lublin, Poland

**Keywords:** addiction, anxiety, depression, experimental animals, model of chronic mild unpredictable stress, nicotine abuse, sex-related differences, stress-related disorders

## Abstract

Nowadays, stressful situations are an unavoidable element of everyday life. Stressors activate a number of complex mental and physiological reactions in the organism, thus affecting the state of health of an individual. Stress is the main risk factor in the development of mental disorders, such as depression and other disorders developing as a result of addiction. Studies indicate that women are twice as likely as men to develop anxiety, depression and therefore addiction, e.g., to nicotine. Even though the data presented is indicative of significant differences between the sexes in the prevalence of these disorders, the majority of preclinical animal models for investigating stress-induced disorders use predominantly male subjects. However, the recent data indicates that this type of studies has also been launched in female rodents. Therefore, conducting research on both sexes allows for a more accurate understanding and assessment of the impact of stress on stress-induced behavioral, peripheral and molecular changes in the body and brain. In this manuscript we have gathered the data from 41 years (from 1981–2022) on the influence of stress on the development of depression and nicotine addiction in both sexes.

## 1. Introduction

Stress can be described as a condition caused by harmful factors and it is characterized by an increase in emotional tension. In addition, it can be defined as a process in which environmental factors (stressors) threaten or even disturb the balance of the body. Under the influence of stress, human organisms have developed various forms of coping with destructive—both external and internal—changes. In other words, stress refers to the need to cope with factors resulting from physicochemical changes in the environment or changes occurring within the internal environment, related to disturbances of systemic homeostasis. Stressors can activate a series of complex mental and physiological reactions in the body, thus affecting the state of health of an individual [1].

The continuous presence of stress in contemporary life made researchers undertake efforts to understand the mechanisms of developing a stress reaction, to explain a tendency to be susceptible to stress and to investigate ways which people use to cope with stress. Much research is now conducted concerning a heavy load of stressogenic factors according to sex, and also into the susceptibility of each sex to developing addiction. As is known, addiction is a chronic, recurring disease of the central nervous system (CNS), characterized by a lack of control over taking an addictive substance [2]. Dependence on psychostimulating agents, including nicotine, is a serious social problem since the mechanisms of its development are not fully understood. Despite the number of ways of treating nicotine addiction, e.g., the use of bupropione, citizine or clonidine, pharmacotherapy still remains unsatisfying [3]. Nicotine, the main component of tobacco smoke, acts through central mechanisms influencing the mood and emotional tension, and contributes to development of physical and mental dependence. These effects, like in the case of other addictive substances, mostly involve dopaminergic neurons in the mesolimbic system, which are a part of the reward system responsible for reinforcing behavior, as for other dependence-causing substances. It is also worth emphasizing that stress is one of the main risk factors in the development of addictions and their recurrence. In the studies conducted by McKee and co-workers, scientists during human research concluded that stress increases HPA axis activation, tobacco craving responses and psychological and emotional reactivity [4]. These studies clearly shows the impact of stress on smoking relapse and continuation of reinforcing the reward-system leading to the development of addiction. As stress influences dopamine levels, it is worth mentioning that there are distinct physiological state pathways of the dopaminergic mesolimbic system according to sex. Thus, it is reasonable to causation of addiction and ways for its treatment through the aspect of stress.

The aim of this article is to review the literature and results from 41 years of studies connected with the relationship between development and severity of the stress reaction and gender, together with the development of nicotine addiction induced by chronic stress in relation to gender. This review is based upon literature data accessible from PubMed concerning the relationship between severity and type of psychosomatic reactions experienced in the conditions of chronic stress in each sex and development of nicotine dependence.

## 2. Regulation of the Hypothalamic-Pituitary-Adrenal Axis in Stress Reactions

Adaptation to stress is a dynamic process which is coordinated by the structures of the CNS. The basic adaptive element of the stress reaction is a stimulation of the hypothalamic-pituitary-adrenal axis (HPA), functioning in close correlation with the central limbic system, and stimulation of the sympathetic-adrenal medullary system (SAM), with its adrenergic and noradrenergic part [5]. The HPA axis and the SAM system cooperate closely with each other, and an increase in activity of one of them causes activation of the other. The relationship between them is particularly visible in the response to chronic stress [6]. The HPA axis, also called the stress axis, is the most important system related to the response of the organism to stress. The HPA axis controls the course of the stress reaction and regulates its activity, thus preparing the body for functioning in a difficult situation [5,7]. The effects of stimulation of the HPA axis appear after several hours and last even for several days. The activity of the HPA axis is regulated by numerous neurotransmitter pathways, which include: GABA-ergic, glutamatergic, serotonergic, noradrenergic and cholinergic. Disturbances in functioning of the HPA axis occur in many mental diseases, such as depression or chronic fatigue syndrome. Due to increased activation of the HPA axis, glucocorticosteroids (GKS) cause damage to the neurons of the hippocampus and frontal cortex, the structures responsible for emotional reactions. A high level of GKS in the blood damages numerous dopaminergic, glutamatergic, serotonergic neurons, and it may also lead to inhibition of the process of neurogenesis. All these changes result in a decrease in the volume of the hippocampus and frontal cortex, which is characteristic of patients with severe depressive disorders [5,8].

Stress is related to the occurrence of a number of physical and mental symptoms, including anxiety and depression. The results obtained in preclinical and clinical examinations suggest that hyperactivation of the HPA accompanying stress reactions is a dominant factor triggering anxiety symptoms and depression (i.e., chronic stimulation of the HPA axis) [9]. A 2003 publication reported that 40–60% of patients with severe depressive disorders have increased cortisol concentrations [10]. In the case of anxiety disturbances a lower cortisol concentration is noted at night and its increased concentration in the morning hours. This indicates that anxiety disorders are related to excessive reactivity to stress reactions, which result from a compensatory decrease in the basic cortisol concentration [11]. Additionally, changes in the HPA axis occurring in people suffering from severe depressive disorders are related to hypersecretion of corticotropin-releasing hormone (CRH)—the hormone also playing the role of a neurotransmitter in the stress reaction [12]. Thus, stress can be considered as the beginning of a pathogenic chain of depressive as well as anxiety disorders [12]. Data concerning concomitant occurrence of depressive and anxiety disturbances indicate that in the clinical picture symptoms of anxiety and depression frequently appear and occur together, and they also complement each other. Scientific research reports that the probability of occurrence of anxiety disorders after an episode of depression is approximately 47–58%; 56% of patients with anxiety disorders develop symptoms of depression [13].

## 3. Neurobiological Mechanisms Underlying Sex-Related Differences in Stress-Related Disorders

The basis of the observed differences related to gender and concerning psychosomatic reactions to stress are both physiological as well as psychological.

Susceptibility to and incidence of many stress-related diseases, including mental, neurological and neurodegenerative disorders, differ substantially in men and women. However, basic neurobiological mechanisms that underlie and determine causes of the differences are still not well-known. This is most probably because of an insufficient number of comparative neurobiological studies which could provide clear-cut, unambiguous data enabling precise evaluation of the changes occurring in both male and female rodents. According to the literature, differences in responding to stress in both sexes often result from the activity of sex hormones, including estrogens, progestogens and androgens. Their activity in the hippocampus plays a significant role in responding to stress.

The literature [14] indicates that neurodevelopmental disorders, such as autism and attention deficit hyperactivity disorder (ADHD), are more frequently observed in men, while women are more likely to develop stress-related disorders, including anxiety or depression, in their adult years of life [15,16]. Moreover, an increased incidence of stress-related disorders is observed in women undergoing intense hormonal changes, i.e., in puberty, pregnancy, postpartum period or menopause [17]. Therefore, interactions between the sex hormones and stress hormones, both in males and females, appear to play a vital role in the development of numerous stress-related disorders.

Significant differences in a reaction to stress of genes in the female and male brains were observed by scientists from the Rockefeller University in New York [18]. Studying differences in the brains, the scientists focused on the CA3 region of the hippocampus, which plays a key role in the response to stress, memory function, mood regulation and processing of information. In the study female and male mice were exposed to stress (6 min of forced swimming) and then the reaction of their neurons and genes in the CA3 region of the hippocampus was examined. The results of the conducted examinations revealed that acute stress affected significantly more genes in the females than in the males (6472 in comparison with 2474). Such a significant difference suggests that a genetic element plays a major role in the response to stress. Genes in the female brains react to environmental stressors to a significantly greater extent than in the case of males. Moreover, while continuing their research, the authors identified 1842 genes reacting to stress in both sexes, which behaved in the opposite way to each other. Additionally, the conducted examinations revealed that the genes activated by stress in females were blocked in males—and vice versa [19].

### 3.1. Assessment of Behavior Related to Anxiety and Depression in Experimental Animals

Information concerning the genetic basis of depressive and anxiety disorders is provided by research conducted on people; however, in order to explain conditions related to these disorders, animal tests for depression and anxiety are applied (see Table 1 and Table 2) [20,21,22,23]. These tests are a useful method which allows us to learn and understand molecular mechanisms involved in pathogenesis of depressive and anxiety disorders, and they are also very useful in searching for and testing new drugs with reference to efficacy of pharmacotherapy applied in the treatment of depression and anxiety in people [23].

### 3.2. Model of Chronic Mild Unpredictable Stress (CMUS)

In order to examine more closely the influence of chronic stress on behavioral effects in rodents, numerous animal models were created, among them the model of chronic mild unpredictable stress (CMUS), which is now one of the most commonly used in experimental rodents (mice, rats). The aim of the model is to induce in examined animals depressive and anxiety behavior, mimicking stressful situations accompanying people’s everyday life. The CMUS procedure was applied for the first time by Willner and colleagues over thirty years ago [24,25].

The mechanism of conduct during the CMUS procedure involves chronic exposure of examined rodents to various social and/or environmental stressors. The used stress stimuli are mild in character, and the frequency of their application each week is changed regularly in order to achieve their maximum differentiation in the performed examinations [26]. The stressors used in the CMUS model include, among others: limited access to food/water, swimming in cold or hot water, changes in temperature, wet litter or a lack of it, squeezing the tail, tilting the cage by 45°, disturbing the day/night cycle, putting rat excreta into mouse cages [27]. The time of action of individual stressors ranges approximately from 2 to 4 weeks [28,29,30].

In the CMUS model, the animals subjected to chronic exposure to the action of mild stressors show behavioral changes resembling symptoms of clinical depression including, among others, a general decrease in sensitivity to reward, impaired locomotor activity or decreased intake of food or water [31]. Additionally, the CMUS model induces anhedonia in the studied animals, which is the basic symptom of depression. Thus, many elements, concerning both pathogenesis of depression as well as its treatment, can be observed based on the analysis of behavior of rodents subjected to the CMUS procedure [32,33].

The CMUS procedure is directly related to etiological factors of depression; in stressed animals it leads to neurobiological, behavioral, physiological and hormonal changes, underlying the psychopathology of changes related to stress [34]. Indeed, the animal CMUS model plays a very significant role in understanding pathophysiological mechanisms of depression, such as changes in the HPA axis or inhibition of neurogenesis, thus decreasing the volume of the hippocampus—the structure responsible for emotional reactions [35,36]. All the behavioral changes induced in rodents in the CMUS model can be reversed by the action of antidepressants [37]. The results obtained so far and the available literature indicate that the CMUS model induces in animals numerous behavioral changes resembling the symptoms of clinical depression in people, such as impaired locomotor activity, reduced intake of food and/or water and decreased reactions to rewarding stimuli [31,38]. However, the available results concerning reactions related to anxiety or depression, induced by earlier exposure of studied animals to stress, are not fully unambiguous.

The majority of stress-related disorders are connected with the aforementioned role of the hippocampus and they result from structural, functional and cognitive changes as well as neurogenic processes occurring in this brain area. Therefore, in order to determine correlations between the effects of sex hormones on the development of stress-related disorders, it seems plausible to determine their effect on the morphology and function of the hippocampus as such. Animal models enable investigation of fundamental neurobiology of sex-related differences occurring in the hippocampus [39]. The function of the hippocampus is closely related to the development of stress-induced disorders as this area of the brain is an important element of the limbic system and is responsible, among other things, for regulation and processing of emotions [40]. Above all else, the hippocampus is a key area of the limbic system accountable for cognitive functions. It is a part of a larger structural network involved in different aspects of learning and memory [41] and its neurogenesis and dendritic morphology have long been associated with learning and hippocampus-dependent memory processes. Getting to know the factors which bring about changes in the morphology and functioning of the hippocampus is undoubtedly pivotal to understanding the development of stress-related disorders. Neuropsychiatric disorders most often result from reduction in the hippocampal volume, which is also characteristic of cognitive deficits [42]. Available research is indicative of substantial behavioral, neurochemical, hormonal and neurobiological differences between the male and female animal models [43,44] According to gathered data, sex hormones contribute to differences arising in the response to stress, both in males and females. Thus, both sex and sex hormones are significant factors that should be taken into account in the etiology of stress-related diseases. Animal studies indicate that the basic adrenocorticotropic hormone (ACTH) level and stress-induced ACTH level as well as the level of corticosterone (CORT) are increased in females in comparison to males [45].

#### Behavioral and Metabolic Changes Induced by Chronic Stress in Experimental Animals

Understanding how stress affects the neurobiological systems linked to mental disorders may lead to elaboration of new therapeutic approaches. Animal stress models provide important information on the neuromechanical mechanisms contributing to the behavioral phenotypes, such as social withdrawal or anhedonia. The literature indicates that stressed animals develop many behavioral and metabolic changes, which may in consequence lead to somatic and mental disturbances. However, the exact neuronal mechanism which is the effect of chronic exposure of studied animals to stress is not yet fully understood. The participation of cholinergic mechanisms in the behavior induced in the CMUS model has also been studied. The expression of acetylcholinesterase in the hippocampus was significantly intensified under the influence of chronic stress [46]; in turn, its activity in the cerebral cortex, hypothalamus and striatum was lowered. Additionally, the density of muscarinic cholinergic receptors in the hippocampus and cerebral cortex was significantly decreased due to chronic exposure to stress [47]. Studies conducted on laboratory animals in the CMUS model revealed that chronically acting stressors cause atrophy of the hippocampus pyramidal cells of the CA3 area and cause a decrease in resistance to other damaging factors (hypoglycemia, hypoxia) [48]. The studies conducted on the subject so far suggest that CMUS induces development of many changes in the cholinergic system, manifested by, among others, disturbances in cognitive functions or depression. This is significant since it has been shown that in the case of severe depression the cholinergic system may become dysfunctional, which is manifested by reduced activity of acetylcholinesterase or increased activity of nicotine cholinergic receptors [49]. However, further research is necessary in order to understand more thoroughly the influence of CMUS on the cholinergic system.

It is worth noting that the bed nucleus of the stria terminalis (BNST) also mediates in the occurrence of sex-dependent differences linked to the behavioral responses to stress. The studies carried out on rodents are still providing valuable information on the neurobiological mechanisms which can also occur in humans. At present, electrophysiological recordings on the human ventral tegmental area (VTA) dopamine neurons are not possible; therefore, examination of the BNST subunits in humans may be performed only with the use of the postmortem tissue samples. However, different modeling systems may be laying foundations for further research. For instance, examination of the social behavior of Cynomolgus monkeys, which present depression-like behavior and whose social system is more similar to the human system, gives a unique insight into how the social stressors affect the brain as well as preservation of species [50]. Thus, conducting numerous studies on rodents and other animals (monkeys, rabbits) appears to bring to light similar characteristics as well as differences among them. The results of such studies may be very helpful in elaborating new therapeutic techniques for the treatment of mental diseases in humans as well.

As for the occurrence of stress-related disorders resulting from the changed hippocampal morphology, sex-related differences in performing hippocampus-dependent functions have been well documented. Men and women differ from one another in their cognitive attributes, social functioning and emotional responses/emotions. Steroid hormones are strong regulators of neurogenesis and dendritic morphology in the hippocampus of the adult since their regulation, among other things, is sex-and age-dependent. Estrogens (17α/β-estradiol, estron), progestins (i.e., progesterone) and androgens (i.e., testosterone) play an important role in the regulation of numerous stages of hippocampal neurogenesis, i.e., proliferation, differentiation and the survival of new neurons during the developmental neurogenesis of the hippocampus in adult men and women [51]. Estrogens exert neuronal and behavioral effects via specific ERα/β estrogen receptors. The effect of estrogens on neurogenesis of the hippocampus and dendritic morphology has been studied observing natural estrogen fluctuations in females, surgical intervention as well as pharmacological intervention. The overall results indicate a positive effect of estrogens on neurogenesis of the hippocampus and dendritic morphology in women. The conducted scientific studies indicate a sex-dependent estrogen effect on hippocampal neurogenesis. Molecular mechanisms mediated by the estrogen effect on this phenomenon in adults may well be ascribed to the activity of estrogen receptors (ERs) ERα and ERβ, which are found in the CA1/3 and DG regions of the hippocampus in men and women [52].

Apart from estrogens, also progestins, such as progesterone, are important regulators which can modulate the effect of estrogen on hippocampal neurogenesis in adult women. Thus, it seems likely that progesterone may antagonize the estrogen effect on cellular proliferation because a single progesterone dose administered 24 h after estrogen administration has been shown to reduce the estrogen-induced increase in the proliferation of cells in female rats. Few presently available studies suggest there is a sex difference in the progesterone effect on neurogenesis. Progesterone, however, increases its effect in men and decreases it in women [53]. Yet, in order to confirm this assumption, more detailed research into this issue needs to be done.

The effect of stress on the hippocampus is complex and appears to be dependent on stress exposure over time as well as its duration. Although striking similarities exist between the animal and human studies concerning the effects of estrogens and androgens, some discrepancies are observed when it comes to the stress effects. While long-term stress may be systematically modeled in animal studies, scientific studies on humans are mainly conducted in clinical populations. Moreover, animal and human studies differ from each other in the way an acute stress is induced and measured, i.e., in humans a more direct approach is applied, while in animals a more delayed measurement of the stress effects is performed. In addition, the studies in humans are focused more on cortisol levels increase as well as on subjective stress assessments in stress responses, which is not entirely possible in animal studies. In spite of these methodological differences, certain similarities may be observed. It appears that in both animal and human studies chronically increased cortisol levels negatively affect the morphology of the hippocampus. Animal studies provide quite convincing evidence that the effects are more intense in males if they occur before puberty, while in females, the effects are more intense if they occur during or after puberty. While animal studies point explicitly to various effects of stress on the hippocampus of males and females, very few human studies have focused on sex differences in relation to the stress effects in the human hippocampus so far. Direct comparisons between males and females, however, are scarce in all species. The research into the role of sex hormones and stress hormones in morphology and functioning of the hippocampus is being continued in numerous preclinical and clinical studies.

## 4. Influence of Stress on Developing Addiction

Stress plays a significant role in the development and course of addictions. Addiction is a deeply rooted response to stress, an attempt to cope with it using “soothing substances”. In people dependent on psychoactive substances, such as narcotics, nicotine or alcohol, it is very easy to trigger the mechanism of the reaction to stress. A hormonal response of the organism to emotions not only overwhelms the strained ability to think rationally, but also increases sensitivity to psychoactive substances. Thus, stress is one of the main causes of permanent dependence on, among others, nicotine. Stress increases a craving for nicotine, strengthens a feeling of satisfaction after nicotine ingestion, thus encouraging further attempts to seek and use it [54]. As the literature data show, stressful experiences increase an individual’s susceptibility to addiction, both at its early stages and in a situation of its recurrence. It was observed in animal studies that stressful stimuli, similarly to addictive agents, intensify the release of dopamine in the mesolimbic system and corpus striatum [55]. Stress reduces the activity of dopamine receptors in neural pathways related to our emotions, which are found in the forebrain, mostly in the nucleus accumbens. When a craving for drugs increases, the functions of dopamine decrease. The mechanism of action of addictive agents is more and more often related to induction of changes which accompany long-term potentiation (LTP). In the studies on experimental animals it was observed that stressful stimuli induce formation of LTP in dopaminergic synapses of the ventral tegmental area (VTA). A significant role in this process seems to be played by changes in glutamatergic transmission. It was demonstrated that stressful stimuli, similarly to addictive substances, alter the ratio of glutamatergic receptors AMPA to NMDA, in favor of AMPA [56].

The studies conducted on animals indicate that stress induced by isolation leads to numerous changes in the brain receptors and increases a tendency to use psychoactive substances in young animals; in adult animals, in turn, it decreases the activity of dopamine-dependent nerve cells. The results of the conducted studies indicate that, in opposition to rats kept in isolation, the animals living together and forming stable groups were not interested in ingestion of an addictive substance, cocaine in this case [57]. Worth mentioning is the fact that prenatal stress induced by the intake of harmful substances by the mother also predispose for the development of addiction in adult life. As was confirmed by Duko etal., 2022, prenatal alcohol and tobacco exposure favors an offspring’s cannabis use [58]. Another evaluation conducted by Nomura and colleagues indicates that matenal smoking during pregnancy is associated with an increased risk of alcohol use disorder in adulthood [59].

### 4.1. Nicotine—A Highly Addictive Substance

Nicotine is the main addictive chemical compound found in cigarettes. In the brain, nicotine binds to and activates nicotine receptors (nAChR). The most common nAChR receptors in the brain are those containing the α4 and β2 subunits, and nAChR, especially those containing the β2 subunits, mediate the nicotine reinforcing action. The β2 subunit plays an important role in the release of dopamine induced by nicotine, in the behavioral response to nicotine, such as nicotine self-administration, conditioned reinforcement, conditioned place preference (CPP) as well as in locomotor activation [60].

The conducted preclinical studies consistently demonstrate that nicotine causes marked ”up-regulation” of the β2 subunit which contains nAChR (β2-nAChRs) in all brain regions [61].

The rewarding action of nicotine, like in the case of other addictive agents, involves mainly dopaminergic neurons in the mesolimbic system, included in the reward system. Activating the dopaminergic mesolimbic system, nicotine causes an increase in locomotor activity and induces seeking behavior, which is related to its enhancing effects [62,63].

Nicotine addiction is a chronic multifactorial disorder characterized by a relatively high rate of relapse even after long period of abstinence. The pharmacological properties of nicotine are complex, and the effects of this drug have been extensively investigated on both humans and animals. Successful smoking cessation is difficult to achieve because nicotine causes both physical and psychological addiction. The nicotine abstinence syndrome in humans is generally unpleasant and includes irritability, anxiety, depressed mood, restlessness, concentration difficulties and nicotine craving [64]. Stress is also one of the key factors in facilitating reward associated with initial and prolonged drug exposure. It has also been revealed that human subjects commonly report stress as the primary cause for their continued nicotine abuse [65].

The relationship between stress and effects of action of nicotine—one of the most often abused psychoactive substances—is not fully understood and coherent [66]. The studies revealed that exposure to stressors increases the volume of inhaled nicotine smoke and intensifies the need to smoke another cigarette [67,68]. It was noted that regular use of psychostimulating agents, including nicotine, is for many people a non-constructive strategy of coping with stress [69].

The literature shows that chronic exposure to nicotine in addicted people may lead to impaired cognitive functions and mood swings with anxiety symptoms and depression [70]. Moreover, certain areas of the brain, in particular the medial prefrontal cortex (mPFC), one of the main tasks of which is to control behavior and adjust it to environmental conditions, become particularly active under the influence of nicotine and act together along with development of memory and anxiety disorders induced by stress [71]. However, there is little information in the literature describing nicotine as a stress-generating factor whose action is associated with disturbances in cognitive functions and mood related to anhedonia [72]. It was shown in some experimental animal models that chronic and short-time stress may aggravate both behavioral and neuronal effects induced by nicotine administration [73]. In other publications, it was noted that nicotine administration in animals abolishes behavioral and neuronal disturbances induced by chronic stress [74,75]. Still, data concerning interactions between nicotine and chronic stress suggest that both nicotine and the CMUS model mutually intensify their negative influence on behavioral functions in studied animals, among others on memory and anxiety disorders [76]. In studies conducted on people, an increase in frequency of smoking in people exposed to everyday stress was observed, with an explanation that this type of dependence decreases their subjective feeling of emotional tension related to a stress factor [68]. The influence of nicotine on depressive behavior, anxiety level and memory processes in animals, in relation to the dose, route of administration and test used has already been described many times [76,77,78]. Information concerning the influence of nicotine on the behavioral effects mentioned beforehand is limited.

Therefore, the issue concerning the influence of nicotine on alleviation of stress in people and animals still remains controversial and not fully understood. One of the main reasons why this subject is still unclear is the lack of detailed scientific data concerning pathways of changes regulating behavioral and metabolic response to stress and/or nicotine. In the preclinical studies, in order to assess the addictive properties of investigated substances, including nicotine, the conditioned place preference test (CPP) is most commonly carried out [79]. This test makes possible the evaluation of the complexity of the mechanisms of action of addictive substances [80]. In order to assess the reinstatement of nicotine place preference, the reinstatement of CPP is commonly applied. Information on the interaction between nicotine and chronic stress suggests that both nicotine and the CMUS model mutually enhance their impact on behavioral functions in the animals [76]. Available literature data indicates that chronic stress intensifies the exploratory behavior associated with the intake of addictive substances, including nicotine. It proves the role of chronic stress on the development of addiction [38]. Stress restores nicotine place preference in the tested animals, which suggests the role of stress in initiating the relapse into addiction after a period of abstinence. Additionally, a number of studies have clearly shown that nicotine has a positive reinforcing effect in the CPP test in rats [81]. The literature points to sex-related differences in the place preference reinstatement induced by stress. Scientific studies have also been conducted to investigate and assess the stress-induced reinstatement of nicotine CPP in male and female mice, and the results showed statistically significant behavioral differences between males and females in response to nicotine. The results of the conducted studies showed that in females the development of nicotine CPP as well as its stress-induced instatement occurs after administration of a higher dose of nicotine (0.75 mg/kg) in comparison to the males in which the development of nicotine CPP is observed at the dose of 0.5 mg/kg [82].

#### 4.1.1. Sex Differences in Nicotine Dependency and Depressive Tendency among Smokers

It should also be emphasized that stress is one of the main risk factors of developing addiction and its recurrence. Addiction is a complex disease of the CNS, characterized by a constant compulsion to seek and administer a psychoactive substance. Dependence on psychostimulating agents, such as nicotine or cannabinoids, and cross dependence are complex mental and neurological disturbances which now pose a big social problem.

Tobacco smoking is one of the major public health problems of the modern times [WHO, 2017]. Despite easy access to medication helpful in giving up smoking, over 70% of smokers who make an effort to quit are back to smoking within a year. According to available data, the likelihood of a relapse is 31% higher in women. Smoking is a major cause of morbidity and mortality in the USA [83]. Smoking prevalence is still higher among men (20.5%) than among women (15.3%) [84] Yet, women are less responsive to the treatment of nicotine dependence. Studies also show that women find it harder to quit smoking in comparison to men, and the currently used methods of treatment, including nicotine replacement therapies (NRT) or treatment with bupropion, are not so effective in women as they are in men [85]. All this emphasizes the importance of investigating the sex differences in the neurobiological mechanisms underlying behaviors connected to smoking so that sex-dependent treatment strategies for nicotine dependence could be optimized.

#### 4.1.2. The Development of Nicotine Addiction in Men vs. Women

Depressive tendencies and nicotine dependence are both factors linked to the lack of success in quitting smoking. In general, women are more susceptible to depression and find it more difficult to give up smoking than men as already stated. However, the relationship resulting from sex differences and their nicotine dependence and susceptibility to developing depression still remains unclear.

Nicotine readily penetrates into the CNS and induces complex central effects depending on both its dose and the circumstances in which it is taken. As a result of the stimulation of presynaptic nAChRs and 5-HT1A receptors, nicotine increases the release of 5-HT. Thus, these effects are responsible for the occurrence of depressive and anxiety effects observed after nicotine withdrawal [86].

The data [87] indicates that worse metabolic parameters are observed in smoking men in comparison to smoking women. The conducted studies show that overweight/obese patients have high cortisol levels in the saliva, which is reflective of their stress level. Therefore, it appears that being overweight/obese is closely connected with psychosocial stress. The ways of stress management are different in men and in women [88,89]. In stressful situations, women tend to seek support from other people [90], while men often resort to grazing or even binge eating [91]. Hypertriglyceridemia observed in men is probably a result of activation of smoking-induced lipoprotein lipase in the adipose tissue [92].

The literature suggestss that twice as many women as men develop depression. During their first visits, smoking women have much higher scores than men in the SDS scale, which is suggestive of depression. The studies conducted so far show that individuals with higher scores in the SDS scale [93] are reportedly more frequent nicotine abusers [94] and less frequently give up this habit for good [95]. In addition, research shows that the withdrawal syndrome is more severe in smokers with higher depressive tendencies in comparison with non-smoking individuals [96]. The factors contributing to the development of depression in smoking women are, among others, frequent hormonal disorders as well as psychological factors, e.g., social stress. The conducted studies point to a greater tendency toward depression in women than in men and indicate that depressive behavior associated with smoking is observed mainly in women [97]. Since women are more susceptible to developing depression than men, additional factors may increase the efficacy of anti-nicotine therapy in women with severe nicotine dependence. Such factors may include the use of antidepressants [98], as an addition to the routine pharmacological therapy, e.g., varenicline or bupropion, as well as cognitive-behavioral therapy as supplementation in the medical treatment [99]. The literature suggests that introduction of therapy for cessation of smoking in the early follicular phase is effective [100]. As stated beforehand, women find it more difficult to quit smoking, and nicotine replacement therapy (NRT) is less effective in women than in men, which is indicative of apparent sex-related differences in the neurochemical mechanisms underlying nicotine-induced behavior. Nevertheless, further investigations are being carried out in order to give grounds for precise evaluation of the differences occurring between smoking women and smoking men.

The available literature also suggests that there is a correlation between the age, tendency to develop nicotine dependence and occurrence of depression symptoms. Older people are generally more susceptible to various diseases, which makes them more likely to develop depression [101,102]. These conclusions were drawn on the basis of a study conducted in advanced-age smoking volunteers with a relatively high level of depressive tendency.

### 4.2. Sex Differences in the Nicotinic Acetylcholine and Dopamine Receptor Systems Underlying Tobacco Smoking Addiction

The studies concerning nicotine-induced male-female differences occurring in the system of acetylcholine nicotine receptors as well as in the dopaminergic system appear to be of great interest [103].

The results of the preclinical studies into this issue have shown that in smokers, after 7–9 days of abstinence, markedly higher levels of β2-nAChR receptors were observed in the cerebral cortex and corpus striatum in comparison to nonsmokers [104]. According to the studies, β2-nAChR is of key importance for the reinforced activity of nicotine. Thus, differences in the β2-nAChR levels between the sexes may be fundamental to understanding behaviors related to smoking both in males and females. So far, only one conducted study has assessed sex-related differences in the levels of β2-nAChR in men and women smokers in comparison to nonsmokers [105]. The results of preclinical studies started to be interlaid and correlated with the results of the studies in humans. Images of the receptors showed that the β2-nAChR availability is much higher in the corpus striatum, cerebral cortex and cerebellum of smoking men in comparison to non- smoking men. In smoking women, however, the level is the same as in non-smoking women. The results of these studies provide a possible neurochemical explanation of the response to NRT depending on the sex of the treated individual. As mentioned before, NRT is a much more effective form of the nicotine dependence treatment in men than in women [106]. This is most probably connected with an increased β2-nAChR expression in men in response to smoking. Clinical studies with participation of tobacco smokers also assessed the mechanisms underlying sex differences in the smoking-regulated effect on the mesolimbic dopaminergic system. The preliminary results showed lower availability of the dopaminergic D2 receptor in the caudate nucleus and the shell of the brain of the smokers addicted to nicotine in comparison to nonsmokers [107]. Dr Edythe London Group [108] confirmed this hypothesis detailing study results which took into account sex-related differences. According to their conducted study results, smoking men showed lower D2/D3 receptors availability in the dorsal striatum (e.g., in the caudate nucleus and the shell of the brain) in comparison to the non-smoking men. Interestingly, smoking women showed similar availability of the DA D2/D3 receptors in comparison with non-smoking women [108]. The results of the studies indicate that chronic tobacco smoking does not decrease the availability of the striatal dopaminergic D2/D3 receptors in women.

Since the striatal dopaminergic neurons arise in the midbrain, Dr London’s group subsequently studied sex differences with respect to the availability of the D2/D3 receptors in the midbrain [109]. The results showed that in smoking women the availability of these midbrain receptors was higher in comparison to non-smoking women. However, the studies did not show any difference in the availability of the D2/D3 midbrain receptors between smoking and non-smoking men. Dr London’s results suggest that higher availability of the D2/D3 midbrain receptors in smoking women may moderate the decrease in the levels of these receptors which was earlier detected in the striatum of smoking men. These studies help to understand the levels of D2/D3 receptors but they fail to point out the function of the dopaminergic system. Innovative PET techniques used in vivo by Morris and Cosgrove in order to investigate dynamic changes in the dopamine (DA) release during smoking indicate that male smokers increase the DA level constantly and fast in the central striatum, while women do not show any increase in the DA level in the central striatum during nicotine use [110]. Bearing in mind the results obtained by London and Cosgrove it appears that tobacco smoking men generally present lower D2 availability and a higher DA release in the striatum, while tobacco smoking women present higher D2 availability in the midbrain and a lower DA release in the striatum.

Smoking women have higher levels of D2 receptors in the midbrain in comparison to both non-smoking women and smoking men [109]. Since the DA neurons originating in the midbrain display an inhibitory activity, it has been hypothesized [108] that higher midbrain D2 receptor availability in smoking women may inhibit the DA release in the central striatum [110], which may in turn weaken the decrease in regulation of the D2 receptors in this region [108]. The study results indicate that the findings in the limbic system may also apply to the mesocortical system. Further studies are currently being conducted to shed more light on this issue. It has been hypothesized that smoking women will present pharmacologically induced lower DA release in PFC in comparison to non-smoking females and smoking males, and that female smokers will not display the same D2 receptor decrease in PFC as smoking men.

Male smokers show a nicotine-induced increase in the β2-nAChR expression as well as smoking-induced DA release in the central striatum, while women do not. In addition to the presented facts that men smoke cigarettes to reinforce the nicotine effect, these results suggest an explanation why therapies targeting nicotine and reward systems are more effective in men than in women. It is worth noting that nicotine may also affect corticotropin-releasing factor (CRF) and the HPA as well as the noradrenergic system in a sexually dimorphic manner during nicotine exposure and its withdrawal. Nevertheless, it is essential that further research into this matter is carried out in order to explain the role of the brain stress pathways in the sex-dependent mechanisms and underlying nicotine dependence. The results of earlier studies showed that in order to reduce negative emotions and stress, women smoke cigarettes more often than men [111]. Women feel the negative effect of giving up smoking much more strongly, and, what is more, susceptibility to relapse under stress is much higher in women [111]. Acute and chronic exposure to stress impairs the PFC function in humans [112], and the decreased/compromised PFC functioning may be one of the mechanisms through which stress triggers a relapse [113], especially in women. Therefore, the stress-induced PFC impairment as well as the changes in the DA signaling associated with it may upset the smokers’ ability to respond properly to negative affect and stressful events, thereby increasing the likelihood of the illusionary feeling that smoking relieves all unpleasant symptoms [114].

### 4.3. The Impact of Oxytocin on Stress and Addiction: The Role of Sex

Another, equally as interesting, issue is the role of oxytocin in response to stress and dependence. More and more studies point to oxytocin as an agent which can affect stress, fear or processing of negative emotional stimuli. Since oxytocin is currently investigated with respect to its application in the treatment of mental disorders characterized by dysregulation in the cerebral stress systems, it is essential to assess if oxytocin is able to effectively reduce certain characteristics associated with stress in both men and women. For example, in the disorders linked to dependencies, it has been observed that oxytocin affects the activity in the circuits connected with the negative reinforcing effect of mind-altering substances [115]. The literature indicates that oxytocin administration compensates some neuroadaptations in the systems of cerebral stress which result from repeated administration and discontinuation/withdrawal of addictive substances; they are believed to contribute to an enhanced response to stress, and negative emotional states as well as higher anxiety connected with acute or extended withdrawal [116]. Thus, oxytocin may counteract the unpleasant emotional states which enhance negative reinforcement to subsequently reduce the compulsive search for addictive substances [117]. The study results show that oxytocin administration may relieve abstinence symptoms, reduce fear and prevent a stress-induced relapse. The similarity of the effect that oxytocin has on this behavior in women and men remains unknown. Bearing in mind the fact that oxytocin may differently affect the activity linked to stress in men and women, similarly to many other aspects of dependence, the negative reinforcement effects may be different in men and in women [118]. Further investigation into whether oxytocin can reduce negative emotional states associated with addiction and whether these effects differ between the sexes is needed.

The necessity to continue scientific studies which would help understand under what conditions oxytocin produces sex-specific effects and explain the effect of sex steroids on these processes remains connected to issues stated above.

To sum up, the molecular mechanism concerning the effects induced by nicotine in both humans and experimental animals subjected to the CMUS procedure is not fully known. It seems significant that one of the main physiological effects caused by nicotine is an increase in activity of the HPA axis, which results in increased secretion of ACTH and GKS [118,119,120]. Moreover, cigarette smoking increases the level of circulating cortisol in humans [121]. Interactions between nicotine, stress and syndromes related to anhedonia result from mutual intensification of negative effects caused by nicotine and CMUS [72,73], through activation of the HPA axis. Data confirms that, despite the different character of the stimulus, stress and nicotine lead to stimulation of the CNS [122,123].

The available literature indicates that in animal experimental models acute stress enhances the rewarding effects of some addictive substances, including psychostimulants and opioids. Traumatic and stressful situations can lead to the development of numerous substance abuse disorders [124,125]. However, limited data in the literature describe common mechanisms related to stress and the development of nicotine addiction. Stress increases DA release in the mesocorticolimbic system [126], and as is known, this neurotransmitter plays a key role in the rewarding effect of nicotine and other addictive substances [127,128]. As a result of the action of acute stressors, the extracellular DA level in NAc (brain pleasure center) is increased [126], which proves the potentiation of dopaminergic transmission under stress. Moreover, enhanced DA release can be sustained for at least 24 h [129,130] which suggests that acute stress exacerbates the rewarding effects of addictive substances such as nicotine. This is of particular interest given that the mesocorticolimbic pathway plays a key role in the development of addictions [127,128]. It has been shown that the acute stress induced by forced swimming enhances the activity of excitatory and weakens the inhibitory synaptic endings of dopaminergic neurons in the midbrain [129,130]. Therefore, it is possible that exposure to stress enhances the development of nicotinic site preference by inducing long-lasting synaptic changes in the reward system dopaminergic pathways.

Nicotine may reinforce the satisfying properties of nonpharmacological stimuli [63,131], and reinforce cognitive functions [132] or exert an effect on stress and anxiety [133], thereby reinforcing behavior related to nicotine dependence. Interactions between different effects of nicotine may be held responsible for well documented interpersonal differences in nicotine use as well as for relapsing to smoking by both humans and rodents [133,134,135]. The data show that women are less sensitive to the primary reinforcing action of nicotine in comparison to men [131], but have a stronger motivation to use nicotine in situations aimed at nullifying stress and negative emotions [136]. This may suggest that stress is a well consolidated, sex-divergent factor contributing to a relapse. The conducted studies also point to significant sex-related differences in the efficacy of medication helpful in giving up the habit. Once more, the literature data indicate that nicotine replacement therapy (NRT) is less effective in women than in men [106]. Other available data, however, suggest that the use of clonidine, an antagonist of the α2-adrenergic receptor, in quitting smoking is more effective in women than in men [137].

## 5. Conclusions

Stress is a reaction of the organism to experiences and events of an unpleasant nature. It is described as severe nervous tension and it either has a mobilizing effect on people or acts destructively. Long-term, chronic stress has a negative impact on health—it contributes to a plethora of diseases, such as cerebral stroke or heart disease. Chronic stress also affects body immunity and decreases concentration, mental stamina and perceptiveness.

The body’s reaction to stressors is a highly complex process that involves the CNS, adrenal and cardiovascular systems. The severity of the stress reaction is related to many factors, including the type of stressful situation, its controllability, the ability to manage stress, etc. In a situation when the balance of the body is threatened, the hypothalamus, situated at the base of the brain, initiates the stress reaction through secretion of a factor releasing CRF. CRF coordinates the stress reaction, inducing a series of interrelated physiological and behavioral reactions. CRF is transported with blood in the brain and subsequently it stimulates the pituitary to produce ACTH. Then ACTH causes secretion of GKS through the adrenal gland, found at the top of the kidneys. GKS, in turn, plays a key role in the course of the stress reaction and its suppression.

Chronic stress is one of the factors which may lead to dependence. Abuse of addictive substances and mood swings induced by chronic stress are very often interlinked. Studies show that people exposed to numerous stressful situations in everyday life frequently abuse different substances, among them nicotine or alcohol. What is more, abuse of substances may often lead to depression, thus triggering a vicious circle of using a substance and subsequent mood swings.

As the studies of Canadian researchers conclude, a factor which greatly determines the reaction to stress is gender. People participating in the study that they conducted were given three tasks related to different levels of perceived stress. During each task the scientists checked the pulse and blood pressure of the participants and measured the level of cortisone in their saliva. Having analyzed the results of the studies which they conducted, they found that the women’s reaction to stress was stronger than men’s, both from the cardiovascular and autonomic nervous system as well as from the hormonal system. However, in order to describe long-term effects of chronic stress, both in the female and male sex, further research in this field is necessary [138].

Women, in comparison to men, are more reactive physiologically and therefore their mobilization for stress is related to stronger stimulation. It has been proved that women differ from men in the range of their hormonal reactions to stress [139]; the observed differences may reflect their different cognitive and emotional transformations of environmental stimuli which, through giving them meaning, initiate other physiological processes. It can be considered that in difficult situations women use such transformative strategies of coping with stress which may perhaps cause a greater physiological burden.

As mentioned earlier, chronic stress is also the factor which creates favorable conditions for the development and course of addiction and its recurrence. Whether a person smokes under the influence of stress depends on a number of factors, including genetic conditioning of smoking as a reaction to stress, behavior of an individual related to smoking, expectations concerning influence of nicotine on stress, severity and type of a stressor, individual feeling of control over a stressful factor, and the range of methods used to manage stress [140]. The differences between the sexes in context of the formation and course of nicotinism are also significant.. Relationships between gender and a risk of nicotinism mostly result from biological and sociological conditioning. Many researchers explain them by the negative influence of the society, for example, by applying pressure to have a successful career, conforming to social norms etc. Another issue concerns biological differences, mostly related to the production of testosterone and estrogen, and also to the average size and composition of the body which causes substances to affect the organism in different ways. A major role is played also by personal predispositions, for example, a tendency to anxiety or depressive reactions, low resistance to stress, low self-esteem. Nevertheless, research aimed at explaining the exact role of stress in inducing nicotinism is being continuously conducted and much remains to be discovered.

## Figures and Tables

**Table 1 brainsci-13-00121-t001:** Review of behavioral tests to assess depression-like behavior.

Type of Test	Test Characteristics	Measured Parameter
Forced swimming test	The apparatus is a cylindrical vessel (25 cm × 10 cm) filled with water up to a height of 10 cm (temperature 23–25 °C).	Measurement of the time of immobility.
Tail suspension test	The test procedure is to tether the mouse by the tail and hang it in the air as it tries to free itself from an unpleasant situation.	Measure the time of active efforts to break free or measure the time of immobility.
Sucrose preference test	The test procedure consists in subjecting the test animals to drinking adaptation of 1% sucrose solution or water twice a week (for a period of 5 weeks). Each single one hour trial is preceded by 14 h of water and food deprivation.	Measure the amount of sucrose solution consumed by weighing the bottles before and after each test as a measure of anhedonia.

**Table 2 brainsci-13-00121-t002:** Review of behavioral tests to assess anxiety-like behavior.

Type of Test	Test Characteristics	Measured Parameter
Elevated plus maze test	The apparatus is a maze consisting of 4 closed and open arms. The maze is set about 50 cm above the ground. This test is performed in an acoustically isolated experimental room lit with low intensity light	(1) Measurement of residence time in open arms. (2) Measurement of the number of entries into open arms. (3) Number of entries to closed arms and total number of entries to both arms.
Light field and dark field test	The apparatus consists of an open room and a closed room that are connected to each other by a tunnel	The number of swings and entrances to the bright room and the time spent in the bright part.
Open field test	The apparatus is a square arena that is available in 100 × 100 × 60 cm, made of white boards. The floor of the arena is divided into central and peripheral squares. A light source is placed above the central part.	(1) Locomotor activity measured on the basis of the number of squares crossed. (2) Number of entrances to central squares and residence time in the central part of the arena. (3) The number of droppings left by the animal.

## Data Availability

Not applicable.

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
