# Peer review of "Recent Studies on the Development of Nicotine Abuse and Behavioral Changes Induced by Chronic Stress Depending on Gender"

_brainsci, 2023, doi:10.3390/brainsci13010121_

Round 1

Reviewer 1 Report

The authors intend to show a state of the art on correlation between chronic stress and nicotine abuse. In my opinion, it is an interesting topic, however, there are several important points that need to be addressed:

Main points

1. There is a lot of unnecessary information about acute animal models of anxiety and depression that are not directly correlated with the topic. However, there is no information on models to assess nicotine abuse, and only one model is mentioned in the manuscript.

2. Animal models are only explained at the beginning of the manuscript, however in the nicotine abuse items, only human studies are reported.

3. The authors focus a lot on sex-dependent differences in different aspects of stress and nicotine abuse. This is an interesting point; however, they do not organize or provide concise information on this.

4. Overall, the manuscript is poorly organized, and the content does not address the topic described in the abstract.

5. The title does not reflect the content of the manuscript, which in my opinion is an important point to show the message described in the manuscript.

6. At some points the authors mention sex-dependent differences in the development of depression, however, information on the correlation between depression and nicotine abuse is missing.

7. The entire manuscript should be revised to clearly develop the different important points needed to explain the available information on the topic.

 Minor points:

1. Some references are missing. It is imperative to revise the entire manuscript to correct these errors (e.g., pages 85, 103, 392, and 464).

2. The grammar should be reviewed by a native speaker to aid in a concise and clear explanation throughout the manuscript.

Author Response

                                                                                                                      Lublin 22.12.2022

Dear Reviewers,

            We would like to thank you for your valuable advice, thanks to which we could improve our manuscript. We have addressed each of your comments and made corrections. For some of Your remarks, we prepared the answers in which we would like to explain our point of view on this manuscript.

Reviewer 1.

Remark 1. “There is a lot of unnecessary information about acute animal models of anxiety and depression that are not directly correlated with the topic. However, there is no information on models to assess nicotine abuse, and only one model is mentioned in the manuscript.”

Thank You for this remark. However, the purpose of the review was not to describe experimental models in nicotine addiction, but only to demonstrate the effects and correlations between stress and nicotine, hence we focused on describing one experimental model. On the other hand, information on animal models of anxiety and depression has been described due to the fact that nicotine also significantly affects the alleviation of many depressive and anxiety behaviors and memory disorders induced by chronic stress in animals. In our opinion, adding a fragment describing animal models of addiction and nicotine addiction itself could significantly increase the volume of the work and deviate from the main goal of this review, which is the stress- and depression-related mechanisms and behavioral changes in humans and animals. Please note: the conditioned place preference test which measure the rewarding effects of drugs has been mentioned.

Remark 2. “Animal models are only explained at the beginning of the manuscript, however in the nicotine abuse items, only human studies are reported.”

  • Thank you for this remark. We added information in chapter 4.1. “Nicotine - a highly addictive substance” (lines 411-423).

Remark 3. “The authors focus a lot on sex-dependent differences in different aspects of stress and nicotine abuse. This is an interesting point; however, they do not organize or provide concise information on this.”

  • Thank you for the suggestion, but it is worth noting that the subject of the article on the correlation of stress and examining nicotine depending on gender is very well researched. In our opinions, we described the issues in a relatively clear and systematic way, and additionally supplemented our review with some information suggested by Reviewers, which may further increase the substantive value of the article.

Remark 4. “Overall, the manuscript is poorly organized, and the content does not address the topic described in the abstract.”

  • Thank you for pointing this out. We added some information in the abstract to specify the purpose of the manuscript.

Remark 5. “The title does not reflect the content of the manuscript, which in my opinion is an important point to show the message described in the manuscript.”

  • Thank You, the title has been changed according to your suggestion.

Remark 6. “At some points the authors mention sex-dependent differences in the development of depression, however, information on the correlation between depression and nicotine abuse is missing.”

  • Thank you for this remark. We added information in chapter 4.1.2 “The development of nicotine addiction in men vs women” 457-460.

Remark 7. “The entire manuscript should be revised to clearly develop the different important points needed to explain the available information on the topic.”

  • Thank You for pointing this out, we added some missing information.

Minor points : 1

“Some references are missing. It is imperative to revise the entire manuscript to correct these errors (e.g., pages 85, 103, 392, and 464).”

  • All missing literature items have been supplemented and fragments indicated by the Reviewer have been corrected.

Minor points 2.

“The grammar should be reviewed by a native speaker to aid in a concise and clear explanation throughout the manuscript.”

  • Thank you for this indication. The manuscript has been reviewed by a native speaker.

Reviewer 2 Report

Thank you for giving me this opportunity to review the manuscript.

The manuscript submitted for publication by Grabowska et al., titled: "Mechanisms of formation and expression of behavioral effects induced by chronic stress and nicotine abuse: short review", is an interesting review that explores the effects of stress on the behavioral, peripheral as well as molecular changes occurring in both sexes, also focusing attention on the abuse of substances, in particular nicotine.

The manuscript addresses a topic that is and will always be in vogue and has been creating debate for a long time.

The manuscript appears to be well written and detailed in topics.

I think that the author must improve the manuscript making a more wide-ranging analysis and thus making the manuscript more interesting to readers.

Minor revision:

1.     The main topic of the manuscript is the mechanism of the stress in both sexes and the its influence on developing addiction, in particular nicotine addiction. However, there is no reference to nicotine addiction in the abstract. I think that the authors must mention it in the abstract to make it clearer.

2.     The introduction is well written and focused, but in my opinion, the bibliography needs to be improved compared to the many concepts to which the authors refer (i.e. see lines 42-47).

I recommend a literature review (see doi: 10.4103/0019-5545.74303; doi: 10.1111/1440-1681.12967; doi: 10.1016/j.bbr.2016.05.006; DOI 10.3390/ijerph19106152; doi: 10.3389/fnbeh.2017.00023; doi: 10.4103/jfcm.jfcm_290_21; doi: 10.3831/KPI.2020.23.001).

3.     The authors do not take prenatal stress into account. In fact, it is well established in the literature that prenatal stress as well as the use of substances of abuse in critical periods of neurodevelopment can induce neuropsychiatric disorders, such as depression and anxiety, and/or vulnerability to the use of other substances of abuse.

I think it's an important topic to get an even bigger picture about the effects of stress on the development of addiction.

I advise the authors to take a cue from these articles: doi: 10.1177/0269881120916135; PMID: 29517191; https://doi.org/10.1016/j.ntt.2022.107064; doi:10.3389/fnbeh.2016.00031 ;https://doi.org/10.3389/fnbeh.2020.00072; DOI 10.3389/fnbeh.2020.00009; https://doi.org/10.1016/j.neubiorev.2005.04.005; doi: 10.15288/jsad.2011.72.199.

Moreover, the impact of oxytocin, which the authors talk about, is a factor closely related to prenatal stress as the oxidative stress induced by tobacco use during pregnancy affects oxytocin fluctuations (DOI: 10.1177/0960327116639363; https://doi.org/10.1016/j.ntt.2008.07.001).

Author Response

Remark 1. “The main topic of the manuscript is the mechanism of the stress in both sexes and the its influence on developing addiction, in particular nicotine addiction. However, there is no reference to nicotine addiction in the abstract. I think that the authors must mention it in the abstract to make it clearer.”

  • Thank you for pointing this out. The information has been added to abstract according to your indications.

Remark 2. “The introduction is well written and focused, but in my opinion, the bibliography needs to be improved compared to the many concepts to which the authors refer (i.e. see lines 42-47). I recommend a literature review (see doi: 10.4103/0019-5545.74303; doi: 10.1111/1440-1681.12967; doi: 10.1016/j.bbr.2016.05.006; DOI 10.3390/ijerph19106152; doi: 10.3389/fnbeh.2017.00023; doi: 10.4103/jfcm.jfcm_290_21; doi: 10.3831/KPI.2020.23.001).”

  • Thank you for this remark and for the literature you suggested. We added information in the introduction.

Remark 3. “The authors do not take prenatal stress into account. In fact, it is well established in the literature that prenatal stress as well as the use of substances of abuse in critical periods of neurodevelopment can induce neuropsychiatric disorders, such as depression and anxiety, and/or vulnerability to the use of other substances of abuse.

I think it's an important topic to get an even bigger picture about the effects of stress on the development of addiction. I advise the authors to take a cue from these articles: doi: 10.1177/0269881120916135; PMID: 29517191; https://doi.org/10.1016/j.ntt.2022.107064; doi:10.3389/fnbeh.2016.00031 ;https://doi.org/10.3389/fnbeh.2020.00072; DOI 10.3389/fnbeh.2020.00009; https://doi.org/10.1016/j.neubiorev.2005.04.005; doi: 10.15288/jsad.2011.72.199.”

  • Thank you, we agree that this topic is compelling. We added information (lines 352-358).

Reviewer 3 Report

Dear authors

I'm not caught up in "Mechanisms of formation and expression of behavioral effects 2 induced by chronic stress and nicotine abuse: a short review" although it discusses an important topic. If it is to be published, there will be a lot of work to be done.

1- After reading the review abstract, you should align the title with the review abstract in order to connect the two. After reading the abstract of your study, I am unable to determine the purpose of your work.

2 - The study needs to be organized and segmented into different sections in order to be presented in a suitable manner.

It is important that the review has specific objectives in order to provide a comprehensive assessment of the topic.

It is recommended that the author include tables that summarize the overall findings of the latest research in the relevant sections.

5- There are no figures in the review describing the mechanisms by which the behavioral effect occurs.

6- A suggestion is made to the authors for reworking this review and rewriting some of the sections so they meet the objectives of this review.

It is necessary for the authors to be able to demonstrate that this review will be unique in some way. For example, in the abstract it is stated that "similar studies have also been conducted on female rodents". In light of this, this review should summarize all studies that have been conducted on male and female rodents, and should be focused on this aspect of the review.

In this case, I will give the authors a second chance to re-do the review and demonstrate to us that they are able to produce a better version of the review.

Author Response

                                                                                                          Lublin 22.12.2022

Dear Reviewers,

            We would like to thank you for your valuable advice, thanks to which we could improve our manuscript. We have addressed each of your comments and made corrections. For some of Your remarks, we prepared the answers in which we would like to explain our point of view on this manuscript.

Reviewer 3.

Remark 1. “After reading the review abstract, you should align the title with the review abstract in order to connect the two. After reading the abstract of your study, I am unable to determine the purpose of your work.”

  • Thank you for pointing this out. We changed the title of the manuscript and abstract to be more consistent.

Remark 2. The study needs to be organized and segmented into different sections in order to be presented in a suitable manner.

  • Thank you for your indications, however, in our opinion, the review has been divided into appropriate sections and in a relatively legible and clear way, i.e.: introduction, specific chapters and subchapters and conclusions, which constitutes a coherent whole. Thus, we tried to correct the layout of the manuscript in accordance with our assumptions and the reviewers' suggestions, so that it reflects the basic goal of the work and is legible, but without excessive division into subsections.

Remark 3. It is important that the review has specific objectives in order to provide a comprehensive assessment of the topic.

  • Thanks for the suggestion, the abstract of the paper has been slightly edited to clearly reflect the purpose of this article.

Remark 4. It is recommended that the author include tables that summarize the overall findings of the latest research in the relevant sections.

  • We would like to notice that Tables are already assigned to particular chapters (see Table 1 and 2). We believe that it is not necessary to include additional tables as all test results are described in the text. therefore, in our opinion,adding tables containing the same information does not bring anything new to the manuscript and may only mislead the reader due to the complexity of the topic.

Remark 5. There are no figures in the review describing the mechanisms by which the behavioral effect occurs.

  • Thank you for pointing this out although, the purpose of the above article is not to describe the mechanisms of behavioral effects, as its title could indicate, but rather to focus on the correlations between the impact of chronic stress, abuse of addictive substances, including nicotine, and their dependence on gender. Therefore, they have not been described, and the earlier title of the article was changed to more precisely describe the issues described and take into account the content of the article.

Remark 6. A suggestion is made to the authors for reworking this review and rewriting some of the sections so they meet the objectives of this review.

  • Thank you for your attention, although the reorganization of the original text would involve a complete change in its structure, which we would prefer to avoid, the review has of course been supplemented taking into account some of the reviewers' suggestions, which in our opinion sufficiently illustrates the purpose of this work.

Round 2

Reviewer 1 Report

The authors do not respond to several points that I highlighted previously, which I consider imperative to increase the quality of the work.  I still consider the manuscript is poorly structured. There is a lot of information about acute models of anxiety and depression, that is not in the field of the review, and very small information about nicotine abuse, which actually is one of the main points of the manuscript.  

Author Response

                                                                                               Lublin 4.01.2022

Response to reviewer 1

Remark: The authors do not respond to several points that I highlighted previously, which I consider imperative to increase the quality of the work.  I still consider the manuscript is poorly structured. There is a lot of information about acute models of anxiety and depression, that is not in the field of the review, and very small information about nicotine abuse, which actually is one of the main points of the manuscript.  

Answer: Thank you so much for the suggestion, we added a few information about nicotine addiction in chapter 4.1. “Nicotine - a highly addictive substance” (lines 371-381).

We appreciate your suggestion however, our vision of the manuscript was to focus on the correlation between nicotine addiction, stress and gender. We did not want to Focus only on the nicotine addiction. Nicotinism is a very well-known and meticulously  described issue, therefore, in the above article, we decided to focus more on its long-term effects, in combination with other factors, e.g. such as chronic stress. The reorganization of the text would basically involve preparing the article from scratch, which in our opinion would not significantly affect its substantive value. In addition, the above form of the article has already been accepted by the other reviewers, therefore, introducing such large changes at this stage could induce further comments from other reviewers, resulting from the change in the original form of the article.

Reviewer 3 Report

Dear authors,

I have seen what you have done and it is really terrific.

I have minor comments which will enhance the big picture of this article.

  1. Include a headline for this review at the beginning to make the reader's life easier.

  1. Again the tile should be changed to " recent studies that the development of nicotine abuse and behavioral changes in- 2 duced by chronic stress depending on gender"

  2. Can you tell me about the timeline of your research? The study you collected covered the period 2010-2022, for example. It should be added to the abstract as well as the introduction.

  3. I recommend adding a picture to summarize the mechanisms by which nicotine causes depression in both genders. The following is a brief summary of the mechanisms behind this review.

  4. You need to add this article since it supports your points to section 4.1.2 "https://doi.org/10.1016/j.jsps.2020.11.001 "

Author Response

Response to reviewer 3.

Dear reviewer,

We are glad that you appreciated our work on this manuscript. We are also grateful for your indications. We prepared answers to each of your remarks.

  1. Include a headline for this review at the beginning to make the reader's life easier.
  • We added headline according to your indications. We agree that headline will make the reader’s life easier.
  1. Again the tile should be changed to " recent studies that the development of nicotine abuse and behavioral changes in- 2 duced by chronic stress depending on gender"
  • Thank you, the title was changed.
  1. Can you tell me about the timeline of your research? The study you collected covered the period 2010-2022, for example. It should be added to the abstract as well as the introduction.
  • The studies covered 41 years. We agree that this information is valuable, and we added it in abstract and introduction as well.
  1. I recommend adding a picture to summarize the mechanisms by which nicotine causes depression in both genders. The following is a brief summary of the mechanisms behind this review.
  • Thank you for your suggestion. However, the most important information on the mechanisms underlying both depression and nicotinism was already included in our manuscript. Additionally, regarding to the fact, that the topic is extremaly interesting and extensive we are planning to write another article exclusively dedicated to these mechanisms.
  1. You need to add this article since it supports your points to section 4.1.2 https://doi.org/10.1016/j.jsps.2020.11.001
  • Thank you for sending this article. We added this according to your indications.
